# AUTOMATED FINE-GRAINED MIXTURE-OF-EXPERTS QUANTIZATION

## ABSTRACT

Mixture of Experts (MoE) enables efficient parameter scaling in large language models by dynamically activating relevant parameter subsets per input token. Compressing MoE models presents unique challenges due to their inherent sparsity. Traditional quantization techniques, which are typically effective for dense models, prove inadequate when applied to MoE architectures. This paper proposes an efficient MoE quantization algorithm. Specifically, we construct a mixed-precision quantization search space encompassing different granularities from expert-level to channel-level. This approach facilitates precise bit-width resource allocation across model components based on their significance and activation frequency. And then, we leverage evolutionary algorithms to efficiently navigate this search space, autonomously identifying optimal quantization configurations. The synergy between adaptive granularity and automated search effectively mitigates the distinctive quantization challenges inherent to MoE models, culminating in a fully automated framework for efficient MoE quantization. Experimental results indicate that our method achieves significant performance improvements across multiple evaluation tasks, with particularly notable results in low-bit quantization scenarios. When applied to the Mixtral-8x7b-v0.1 model, our approach outperforms the current state-of-the-art by $9.24\%$, setting a new benchmark in MoE quantization. Code is available in supplementary materials.

## 1 INTRODUCTION

In recent years, Large Language Models (LLMs) (Touvron et al. (2023a;b); Reid et al. (2024); Zhang et al. (2022)) have demonstrated remarkable progress in the field of natural language processing. However, the continuous expansion of model scale has concomitantly led to substantial computational resource requirements. The Mixture of Experts (MoE) (Jacobs et al. (1991); Fedus et al. (2022); Lepikhin et al. (2020)) architecture has emerged as an innovative paradigm in large language models, offering efficient parameter scaling through dynamic activation of input-relevant parameter subsets. By partitioning the parameter space into multiple expert networks and employing dynamic routing, MoE models facilitate significant performance enhancements while maintaining relatively low training costs and only marginally increasing computational demands during inference. (Jiang et al. (2024); Dai et al. (2024)) This approach enables the handling of increasingly complex tasks while optimizing resource utilization, representing a significant advancement in scaling model capabilities in natural language processing(Rajbhandari et al. (2022); Chen et al. (2022b)).

As large language models continue to grow in size, the importance of model compression becomes increasingly evident. such as quantization (Frantar et al. (2023); Yuan et al. (2023); Lin et al. (2023); Shao et al. (2023); Ma et al. (2024)), sparsification (Frantar & Alistarh (2023); Sun et al. (2023); Zhang et al. (2023)), and knowledge distillation (Hsieh et al. (2023); Gu et al. (2024)), have demonstrated significant progress in reducing model size and computational requirements while maintaining performance. However, existing compression methods for large models do not necessarily yield the same effectiveness when applied to Mixture of Experts (MoE) models. The inherent sparsity characteristics of MoE models present unique challenges for compression Chowdhury et al. (2023). The dynamic routing mechanism in MoE models refers to the process where input tokens are selectively directed to specific experts based on their content. Shazeer et al. (2017) This leads to significant variations in the activation frequency of different experts, as some experts may be called

upon more frequently than others depending on the input distribution. The heterogeneous expert structure indicates that each expert within the MoE model may have distinct parameters and specialized functions. Unlike traditional homogeneous neural networks, this heterogeneity implies that each expert may require a distinct compression strategy, further complicating the compression process. Moreover, the sparse computational patterns in MoE models make it difficult to directly apply conventional compression techniques without impacting model performance Li et al. (2023).

While quantization has proven effective and widely adopted in dense model compression, research on MoE model compression has predominantly focused on model pruning, particularly expert and layer pruning techniques. Lu et al. (2024) proposed an expert-level pruning method based on router norm changes, retaining experts with the largest changes in router weights. Chowdhury et al. (2024) introduced a theoretically-driven approach, demonstrating that prioritizing the removal of experts with smaller L2 norm changes in their routers after pre-training can guarantee test accuracy under certain conditions. Some researchers have recognized the need for more comprehensive compression strategies. For instance, He et al. (2024) proposed a unified MoE compression framework that combines expert slimming (including pruning and quantization) with expert pruning (such as layer and block deletion). This approach represents a step towards integrating multiple compression techniques. However, a common limitation of these methods is their static nature, where uniform compression techniques are applied across all experts. This approach fails to account for the dynamic activation patterns and heterogeneous structure inherent in MoE architectures, potentially leading to suboptimal compression results. The realm of MoE quantization, in particular, remains largely unexplored. Li et al. (2024) investigated the assignment of different quantization bit-widths based on importance judgments at various granularities of MoE structures. Through extensive experiments to select importance metrics, they pushed compression rates to higher levels. However, the field still lacks a comprehensive quantization solution specifically tailored to the intricacies of MoE models.

In response to the unique challenges posed by quantizing MoE models (Jiang et al. (2024); Li et al. (2024)), this paper introduces an innovative and efficient automated quantization algorithm. Our approach addresses this challenge by constructing a mixed-precision quantization search space that encompasses a wide range of granularities, from expert-level to channel-level. This design allows for dynamic and flexible quantization strategies that can adapt to the varying activation patterns and heterogeneous structure of MoE models. By incorporating different levels of granularity, our method can better align with the dynamic routing mechanism and diverse expert utilization inherent in MoE architectures. Consequently, we extend the granularity of quantization configurations to the internal structures of experts, enabling our search space to capture the dynamic characteristics of MoE models. This approach allows for efficient exploration of the vast search space, minimizing quantization model loss under target compression rates. Ultimately, we propose a fully automated, efficient quantization framework for MoE models that addresses their inherent dynamic nature, providing a more tailored and effective approach than previous static methods. In conclusion, the key contributions of this paper are as follows:

- We introduce a novel mixed-precision quantization search space that achieves fine-grained quantization ranging from expert-level to neuron-level granularity. By extending quantization configurations to the internal structures of experts, our approach not only captures the multi-layered characteristics of MoE models but also flexibly integrates with various importance judgment factors. This integration enables precise allocation of quantization resources across different structural levels of the model.

- Furthermore, to navigate the complex search space efficiently, we introduce an evolutionary algorithm-based approach for identifying optimal quantization configurations. This method enables effective exploration of the vast search space, automatically determining the most suitable quantization strategy while minimizing model loss under specified compression rates. The evolutionary algorithm's ability to handle multi-objective optimization makes it particularly well-suited for balancing the trade-offs inherent in MoE model quantization.

- Our novel techniques have established new benchmarks in the domain of Mixture-of-Experts (MoE) quantization. In the realm of mixed-precision quantization, our proposed method demonstrates remarkable superiority, surpassing existing linear-level quantization approaches by a substantial 7.9% on the neuron-quantized Mixtral-8x7B-v0.1 model, while maintaining identical quantization configurations. Through the innovative integration of evolutionary algorithms, we further enhanced our method's efficacy, achieving an addi-

tional 1.34% performance improvement. A key strength of our approach lies in its orthogonality to existing LLM quantization methods. This characteristic opens up exciting possibilities for synergistic combinations, potentially yielding even more significant performance enhancements.

## 2 RELATED WORK

### 2.1 MIXTURE-OF-EXPERTS MODELS

Mixture-of-Experts (MoE) architectures have emerged as a significant innovation in Large Language Models (LLMs) (Jiang et al. (2024); Dai et al. (2024)), offering a balance between model capacity and computational efficiency. Originally proposed by Jacobs et al. (1991), MoE has evolved substantially in the context of deep learning and natural language processing. Shazeer et al. (2017) pioneered the use of MoE in transformer-based models, demonstrating improved performance on translation tasks. This work was extended by Lepikhin et al. (2020) with the GShard architecture, scaling MoE to trillion-parameter models. Fedus et al. (2022) introduced the Switch Transformer, implementing a sparse gating mechanism for more efficient routing of inputs to experts. The theoretical understanding of MoE models has also progressed.Chen et al. (2022b) provided insights into the generalization capabilities of MoE models, while Chowdhery et al. (2022) offered theoretical guarantees on the computational efficiency of modern MoE architectures with patch-level routing. Recently, the Mixtral model by Jiang et al. (2024) demonstrated that MoE can match the performance of full-parameter LLMs while using significantly fewer active parameters. This has sparked interest in MoE compression techniques. Du et al. (2019) proposed expert-choice routing to optimize expert utilization, while Rajbhandari et al. (2022) addressed the memory overhead associated with storing multiple expert networks. These advancements highlight the potential of MoE for efficient, large-scale language modeling and set the stage for further research in MoE compression (Li et al. (2023); Chen et al. (2022a)) and optimization.

### 2.2 POST-TRAINING QUANTIZATION

Post-training quantization (PTQ) (Wei et al. (2022b); Yao et al. (2022); Ashkboos et al. (2024); Liu et al. (2024)) has emerged as an efficient technique for model compression, particularly beneficial for LLMs. Unlike quantization-aware training or fine-tuning (Tailor et al. (2020); Ding et al. (2022) ), PTQ operates on pre-trained models without extensive retraining (Liu et al. (2021); Fang et al. (2020)). In computer vision, AdaRound Nagel et al. (2020) optimizes weight rounding strategies, BRECQ Li et al. (2021) introduces block-wise reconstruction, and QDROP Wei et al. (2022a) enhances performance through activation substitution. For LLMs, GPTQFrantar et al. (2023) uses approximate second-order information for layerwise quantization, SmoothQuant Xiao et al. (2023) tackles activation outliers, and AWQ Lin et al. (2023) preserves critical weights' precision. OmniQuant Shao et al. (2023) integrates multiple strategies, combining mixed-precision quantization, outlier handling, and adaptive rounding. AffineQuant Ma et al. (2024) introduces an affine transformation to adjust weight distribution, effectively reducing quantization errors. These advancements have significantly improved LLM deployment efficiency on resource-constrained devices (Kim et al. (2023); Chen et al. (2024)).

## 3 METHOD

### 3.1 PRELIMINARY

**The Mixture-of-Experts Architecture**. In Mixture-of-Experts (MoE) models based on the Transformer architecture (Fedus et al. (2022); Jiang et al. (2024)), the traditional dense Feed-Forward Network (FFN) sublayer is replaced by an MoE layer comprising $N$ FFN layers and a router $\boldsymbol{G}$ Shazeer et al. (2017) . These $N$ FFN layers are also called experts$\{\boldsymbol{E}_1, \boldsymbol{E}_2, \ldots, \boldsymbol{E}_n\}$, Given $x \in \mathbb{R}$, the output $y \in \mathbb{R}$ is a weighted sum of the selected experts' output :

$$y = \sum_{i \in K} \boldsymbol{G}_i(x) \cdot E_i(x), \tag{1}$$

where $K$ denotes the indices of selected experts, $\boldsymbol{G}(x) \in \mathbb{R}$ is the gating score vector computed by the router for all $n$ experts based on the input $x$. The router employs a Top-$K$ operation to select the $K$ experts with the highest gating scores. Subsequently, the input is routed only to these selected experts, resulting in a sparse activation pattern. The computation of $K$ is given by:

$$K = \text{TopK}(\text{Softmax}(\mathbf{G}(x))). \tag{2}$$

The expert structure in Mixtral-8x7B-v0.1 Jiang et al. (2024) follows the design of LLaMA Touvron et al. (2023a). Each expert comprises three layers, defined as:

$$\text{Expert}(x) = W_{\text{down}}(W_{\text{up}}x \odot \text{Act}(W_{\text{gate}}x)), \tag{3}$$

where $\odot$ denotes element-wise multiplication and Act represents the activation function. The expert structure involves three weight matrices: $W_{\text{up}}, W_{\text{gate}} \in \mathbb{R}^{d_{\text{mid}} \times d_{\text{in}}}$ and $W_{\text{down}} \in \mathbb{R}^{d_{\text{out}} \times d_{\text{mid}}}$.

This MoE architecture allows each expert to function as an independent FFN module, significantly enhancing the model's capacity without a corresponding increase in computational cost. However, due to the replication of FFN layers, the primary memory overhead and the increase in model size in MoE models are attributed to the FFN components.

**Model Quantization**. Quantization is an effective model compression technique that reduces model size and computational complexity by converting high-precision floating-point parameters and activations to lower-bit representations. This method significantly decreases the storage requirements and computational overhead while striving to maintain model performance. The quantization process Nagel et al. (2021) can be generally formulated as follows:

$$Q(x) = s \cdot (\text{clamp}([x/s] + zp, 0, 2^n - 1) - zp) \tag{4}$$

where $x$ represents the original floating-point value to be quantized, $s$ is the quantization step size, $zp$ is the zero point, $n$ is the target bit-width, $\text{clamp}(\cdot)$ is a function that restricts values to a specified range. $s$ defines the interval between quantized levels, directly affecting precision, while $zp$ adjusts the quantization range to effectively map quantized values to the original floating-point distribution. These two parameters can be computed using the following formulas:

$$s = \frac{\max(x) - \min(x)}{2^n - 1}, \quad zp = \left\lfloor \frac{-\min(x)}{s} \right\rceil \tag{5}$$

To minimize precision loss in large-scale models, GPTQ Frantar et al. (2023) introduces a layerwise quantization technique based on approximate second-order information. This method quantizes weights column by column, subsequently updating the remaining weights based on the OBQ Frantar & Alistarh (2022) approach until all weights are quantized. The update process for the weight matrix $\boldsymbol{W}$ is formulated as follows:

$$\begin{cases} \hat{\boldsymbol{W}} \leftarrow \boldsymbol{W} - \delta_p \\ \\ \delta_p = -\dfrac{\boldsymbol{W}_p - Q(\boldsymbol{W}_p)}{[\mathbf{H}^{-1}]_p} \cdot [\mathbf{H}^{-1}]_{:,p} \\ \\ \varepsilon_p = \dfrac{(\boldsymbol{W}_p - Q(\boldsymbol{W}_p))^2}{2[\mathbf{H}^{-1}]_{pp}} \end{cases}, \tag{6}$$

where the Hessian matrix $\mathbf{H} = 2\boldsymbol{X}\boldsymbol{X}^T$, and $[\mathbf{H}^{-1}]_{:,p}$ denotes the $p$-th column of the inverse Hessian matrix. $\delta_p$ represents the quantization compensation generated when quantizing the $p$-th column, while $\varepsilon_p$ denotes the quantization error.

Outliers in large language models are a critical factor affecting quantization performance (Wei et al. (2022b); Lin et al. (2023); Ma et al. (2024); Xiao et al. (2023)). To mitigate the issue of excessive rounding errors for numerous smaller, non-outlier values caused by an overly large scale due to outliers, an effective approach Shao et al. (2023) is to introduce learnable parameters $\gamma$ and $\beta$ to adjust the maximum and minimum values. Through gradient descent optimization, the optimal truncation range can be determined using only a small calibration set:

$$s = \frac{\gamma \max(\boldsymbol{W}) - \beta \min(\boldsymbol{W})}{2^n - 1}, \quad z = -\left\lfloor \frac{\beta \min(\boldsymbol{W})}{s} \right\rceil \tag{7}$$

Mixed-precision quantization allocate different quantization bit-widths to various layers or modules within a neural network. HAWQ Dong et al. (2019) is a representative method in mixed-precision quantization. It utilizes the ratio of the maximum eigenvalue of the Hessian matrix to the parameter count as a quantization sensitivity indicator, which guides the allocation of appropriate quantization precision to different modules. Existing mixed-precision quantization methods for MoE models (Li et al. (2024); He et al. (2024)) have explored strategies that allocate more bits to frequently utilized experts, initial MoE blocks, or shared experts. While these approaches have demonstrated some improvement in quantization efficiency, they overlook the finer-grained structural features within MoE models. In the following section, we introduce a novel, more granular mixed-precision quantization method. Our approach delves into the internal expert structures, considering multiple levels of granularity from expert-level to neuron-level, thereby achieving more precise and efficient allocation of quantization resources.

### 3.2 NEURON QUANTIZATION

For a MoE expert, a neuron Lo et al. (2024) is defined as the combination of corresponding row vectors from $W_{\text{up}}$ and $W_{\text{gate}}$, along with the corresponding column vector from $W_{\text{down}}$. Consequently, each expert contains $d_{\text{mid}}$ neurons. Based on the definition in Equation 3, the $i$-th neuron of the $e$-th expert in a given Transformer block can be defined as follows:

$$\text{Neuron}_i^{(e)}(x) = W_{\text{down}[:,i]}^{(e)} \cdot (W_{\text{up}}^{(e)}{}_{[i,:]}x \odot \text{Act}(W_{\text{gate}}^{(e)}{}_{[i,:]}x)), \tag{8}$$

where $e \in 1, 2, ..., N$ and $i \in 1, 2, ..., d_{\text{mid}}$. Previous research (Qiu et al. (2024); Geva et al. (2020)) has shown that the projection matrices of experts can be viewed as a key-value system: the column vectors of $W_{\text{down}}$ represent potential outputs; the row vectors of $W_{\text{up}}$ generate weights for each possible output; and the row vectors of $W_{\text{gate}}$ determine the activation of corresponding channels. Consequently, the neuron structure can be considered as a channel-level micro-expert. This perspective allows us to capture the computational characteristics of MoE models at a finer granularity, providing a foundation for more precise optimization strategies.

It is important to note that the sparse activation nature of MoE models implies significant variations in the importance and utilization frequency of different experts. Similarly, neuron-level "micro-experts" face comparable imbalanced activation issues. Current quantization methods for MoE models Li et al. (2024) typically allocate resources at the expert or linear layer level. These coarse-grained approaches may not fully exploit the structural properties of MoE models, potentially overlooking intricate differences within experts and leading to information loss. In contrast, neuron-level quantization offers a more refined and flexible compression strategy. This approach enables better adaptation to the inherent imbalances in MoE models by capturing the importance of each computational unit within the MoE structure. As a result, we can allocate more resources to frequently activated neurons or those with greater influence on the output, thereby improving overall quantization efficiency.

To ensure consistent and precise intra-neuron computations, uniform quantization bit-width is applied to each neuron Li et al. (2021). We define a quantization configuration vector $\mathbf{b} = [b_1, b_2, ..., b_{d_{\text{mid}}}]$, where $b_i$ represents the quantization bit-width for the $i$-th neuron. Specifically, for per-channel quantization, quantization parameters (scale and zero point) are collected along the row dimension for $W_{\text{up}}$ and $W_{\text{gate}}$, while utilizing the column dimension for $W_{\text{down}}$. The quantization parameters for the original floating-point value $x$ to be quantized are computed as follows:

$$\begin{cases} s_i = \dfrac{\max(x_{[i,:]}) - \min(x_{[i,:]})}{2^{b_i} - 1}, & zp_i = \left\lfloor \dfrac{-\min(x_{[i,:]})}{s_i} \right\rceil, & \text{for } W_{\text{gate}} \text{ and } W_{\text{up}} \\[4mm] s_i = \dfrac{\max(x_{[:,i]}) - \min(x_{[:,i]})}{2^{b_i} - 1}, & zp_i = \left\lfloor \dfrac{-\min(x_{[:,i]})}{s_i} \right\rceil, & \text{for } W_{\text{down}} \end{cases} \tag{9}$$

The quantization parameters for the entire weight matrix can be represented as the cumulative sum of the individual neuron parameters:

$$s^{\text{neuron}} = s_i \mid i \in 1, \ldots, d_{\text{mid}}, \quad zp^{\text{neuron}} = zp_i \mid i \in 1, \ldots, d_{\text{mid}} \tag{10}$$

Based on the previously defined quantization parametersd and quantization configuration vector $\mathbf{b}$, the quantization function can be expressed as:

$$Q(x) = s^{\text{neuron}} \cdot (\text{clamp}([x/s^{\text{neuron}}] + zp^{\text{neuron}}, 0, 2^{\mathbf{b}} - 1) - zp^{\text{neuron}}) \tag{11}$$

## 3.3 EVOLUTIONARY ALGORITHMS

Existing quantization configuration allocation methods Li et al. (2024) primarily rely on the inherent characteristics of MoE modules, such as expert utilization frequency or weight magnitude metrics. However, these methods exhibit varying performance across different scenarios, making it challenging to determine a universally optimal strategy. Faced with such a complex decision space, evolutionary algorithms demonstrate significant advantages (Real et al. (2019); Guo et al. (2020); Li et al. (2021)). Evolutionary algorithms Wang et al. (2019) can effectively explore a vast and diverse configuration space, optimizing quantization strategies by simulating natural selection processes. Their adaptability enables dynamic strategy adjustment during the search process, effectively avoiding local optima. This approach can automatically identify the most suitable quantization configuration for specific MoE neurons without presuming the superiority of any single importance metric. Given a fixed total model compression rate $\mathbf{Bit}$, We define the optimization objective of the evaluation function as follows:

$$\min_{c} \sum_{i=1}^{l} \sum_{j=1}^{N} L(\hat{E}_{ij}, c_{ij}), \quad s.t. \quad H(c) = \mathbf{Bit}, c_{ij} \in \{c_1, c_2, ..., c_m\} \quad , \tag{12}$$

where $L(\hat{E}, c)$ represents the loss of an expert under quantization configuration $c$. $l$ denotes the number of layers in the model, $N$ is the number of experts per layer. $c(i, j)$ returns the quantization bit-width for the $j$-th expert in the $i$-th layer. The set $\{c_1, c_2, ..., c_m\}$ is a predefined collection of candidate quantization bit-widths. $H(\cdot)$ is defined as the total bit budget function. The evaluation function returns the negative value of the cumulative error.

$L(\hat{E}, c)$ is computed based on the quantization loss of each weight, utilizing the method proposed in GPTQ. For each expert in the MoE model, the total loss is the sum of the $\varepsilon$ from three linear layers ($W_{up}$, $W_{gate}$, and $W_{down}$):

$$L(\hat{E}_{ij}, c_{ij}) = \sum_{W_{up}, W_{gate}, W_{down}} \varepsilon_{c_{ij}} \quad , \tag{13}$$

where $\varepsilon$ is defined as in Equation 6. In our experimental setup, we define the set of candidate quantization bit-widths as $\{2, 2.5, 3, 3.5, 4\}$. Specifically, 2 represents 2-bit quantization for all neurons in the expert, 2.5 indicates 4-bit quantization for 25% of neurons and 2-bit quantization for the remaining 75%, and so forth, with 4 representing 4-bit quantization for all neurons. When the quantization configuration is set to 2.5, we utilize the magnitude of weight outliers as a selection criterion to determine which 25% of neurons are allocated 4-bit quantization. The choice of 2-bit and 4-bit quantization is motivated by their prevalent use in practical deployments, offering a balance between model compression and computational efficiency.

In the process of accumulating errors for the evaluation functionLi et al. (2021), real-time computation of $L(\hat{E}, c)$ during the quantization process is impractical. This limitation primarily stems from the use of large language models with billions of parameters. Conducting comprehensive quantization and error calculations for each potential quantization configuration would result in exponential growth of computation time, rendering the optimization process prohibitively slow. Moreover, real-time quantization and evaluation of each configuration demand substantial computational resources, posing significant challenges even in high-performance computing environments. Additionally, the quantization process exhibits a cascading effect across transformer blocksFrantar et al. (2023). The computation of the Hessian matrix $\mathbf{H}$ for a given linear layer depends on the output of the preceding block as input. Consequently, layers positioned later in the network tend to accumulate higher errors. This phenomenon introduces a bias in evolutionary algorithms, which tend to allocate higher bit-widths to the later layers to mitigate the cumulative error propagation.

To address the aforementioned challenges, we employ a pre-computation error method. During this process, we individually assess each expert within every transformer block of the model. Specifically, we maintain 32-bit floating-point precision for all layers preceding the target block, while

---

**Algorithm 1** Evolutionary Mixed-Precision Quantization for MoE (EMQ-MoE)

---

**Require:** Population size $P$, max generations $G$, mutation rate $\mu$, number of experts $N_e$, elitism rate $r_e$, tournament size $t_s$, top individuals rate $r_t$, target bit budget $B$
**Ensure:** Best mixed precision quantization configuration
1: **Initialize** population $P_0$ with $P$ individuals, each having $N_e$ balanced genes
2: $E \leftarrow Precompute\_errors()$          $\triangleright$ Precompute errors for each expert
3: $best\_individuals \leftarrow []$
4: **for** $g = 1$ to $G$ **do**
5:      $Evaluate\_population(P_{g-1}, E)$
6:      Sort $P_{g-1}$ by fitness (descending order)
7:      $best\_individuals \leftarrow Select\_top\_individuals(P_{g-1}, r_t)$
8:      $new\_population \leftarrow Select\_elite\_individuals(P_{g-1}, r_e)$        $\triangleright$ Elitism
9:      **while** $|new\_population| < P$ **do**
10:         $parent1 \leftarrow Tournament\_selection(P_{g-1}, t_s)$
11:         $parent2 \leftarrow Tournament\_selection(P_{g-1}, t_s)$
12:         $child \leftarrow Crossover(parent1, parent2)$
13:         $Mutate(child, \mu)$
14:         $Balance\_genes(child, B)$        $\triangleright$ Adjust quantization to meet target bit budget
15:         $new\_population \leftarrow new\_population \cup \{child\}$
16:      **end while**
17:      $P_g \leftarrow new\_population$
18: **end for**
19: Get the best fitted entry from $best\_individuals$
20: **return** Best mixed precision quantization configuration

---

applying various quantization configurations solely to the expert under evaluation. We compute and store quantization errors separately for each expert's linear layers. This pre-computation strategy significantly enhances the efficiency of the evolutionary algorithm and provides a more equitable and accurate foundation for error estimation. Building upon this pre-computation error method, we design an evolutionary algorithm that leverages these pre-computed errors to efficiently search for an optimal mixed-precision quantization configuration. Our algorithm presented in Algorithm 1, synthesizes principles from genetic algorithms with domain-specific optimizations tailored for mixed-precision quantization in MoE models. It leverages evolutionary mechanisms to iteratively refine quantization schemes, incorporating problem-specific heuristics to enhance convergence towards optimal configurations.

Table 1: Zero-Shot Task Performance of Mixtral-8x7B using Automated Fine-Grained MoE Quantization with GPTQ

| Model | Bits | Granularity Level | Method | Accuracy (%) ↑ | | | | | |
|---|---|---|---|---|---|---|---|---|---|
| | | | | PIQA | HellaSwag | WinoGrande | OBQA | COPA | Avg. |
| Mixtral-8x7B | fp16 | - | - | 82.54 | 83.99 | 76.32 | 46.80 | 93.00 | 76.53 |
| | 4 bit | Static | | 81.28 | 81.46 | 76.09 | 46.40 | 90.00 | 75.05 |
| | 3 bit | | | 80.69 | 81.02 | 75.93 | 44.20 | 91.00 | 74.56 |
| | 2 bit | | | 64.15 | 48.53 | 52.01 | 30.80 | 74.00 | 53.90 |
| | 2.54 bit | Expert | Random | 60.14 | 46.60 | 58.59 | 33.00 | 68.00 | 52.26 |
| | | | Frequency | 66.21 | 56.62 | 58.33 | 32.00 | 76.00 | 57.83 |
| | 2.54 bit | Linear | Random | 68.19 | 57.36 | 60.74 | 35,80 | 78.67 | 60.15 |
| | | | Weight Outlier | 68.23 | 57.04 | 62.19 | 35.80 | 83.00 | 61.25 |
| | 2.54 bit | Neuron (Ours) | Weight Outlier | 76.12 | 71.60 | 68.27 | 40.80 | 89.00 | 69.15 |
| | | | evolutionary | **76.55** | **71.60** | **69.69** | **43.60** | **91.00** | **70.49** |

## 4 EXPERIMENTS

### 4.1 SETTINGS

**Implementation Details.** The implementation details of our proposed method are as follows. In alignment with prevailing quantization methodologies for large language models (Frantar et al. (2023); Lin et al. (2023); Shao et al. (2023); Ma et al. (2024)), we employ a calibration set consisting of 128 samples, each comprising 2048 tokens, extracted from the WikiText2 Merity et al. (2016) training corpus. Our quantization strategy implements asymmetric group quantization with a group size of 128. We apply 4-bit precision quantization to all attention-related layers while maintaining the router in its full-precision state. The evolutionary algorithm's hyperparameters are configured as follows: a population size of 400 individuals is maintained across a maximum of 300 generations. We implement a mutation rate of 0.07 and employ an elitism strategy, preserving the top 7 individuals in each generation. The tournament selection process utilizes a tournament size of 5. To ensure consistency across all experimental conditions, we constrain the overall model bit-width to 2.54 bits Li et al. (2024).

**Baseline.** All experiments were conducted on the Mixtral-8x7B-v0.1 model Jiang et al. (2024)). Our method successfully quantizes this multi-billion parameter model, with all GPTQ Frantar et al. (2023) experiments completed on a single NVIDIA RTX 3090 GPU with 24GB memory and all OmniQuant Shao et al. (2023) experiments on a single NVIDIA A800 GPU with 80GB memory, demonstrating its efficiency and scalability. We compare our approach against baselines Li et al. (2024) including expert-level random allocation, frequency-based allocation, and other quantization parameter distribution methods.

**Evaluation.** To assess the efficacy of our automated fine-grained quantization approach for MoE models, we conducted evaluations across five zero-shot tasks: PIQA Bisk et al. (2020), HellaSwag Zellers et al. (2019), WinoGrande Sakaguchi et al. (2019), COPA Gordon et al. (2012), and OpenbookQA Mihaylov et al. (2018). We utilized the lm-eval-harness Gao et al. (2021) framework to obtain individual task accuracies and the overall average accuracy. This comprehensive evaluation strategy enables us to gauge the impact of our quantization method on various aspects of model performance, providing insights into its generalization capabilities across diverse natural language understanding tasks.

### 4.2 RESULT

Table 1 presents the results of our evaluation of the Mixtral-8x7B-v0.1 model Jiang et al. (2024) using our proposed automatic fine-grained MoE quantization method with GPTQ Frantar et al. (2023) across multiple zero-shot tasks. The "Granularity Level" column indicates the level at which quantization bits are allocated in mixed-precision quantization, while the "Method" column denotes the bit allocation strategy. In the study by Li et al. (2024), the expert-level "Frequency" method allocates 4 bits to the top 25% most frequently used experts in each Transformer block, with the remaining experts receiving 2 bits. The linear-level "Weight Outlier" method quantizes the 25% linear layers with the highest weight anomalies across all $W_{up}$, $W_{gate}$, and $W_{down}$ layers in the entire model to 4 bits, leaving the rest at 2 bits. Our method demonstrates significant improvements. When the granularity level is refined to neuron-level quantization, the "Weight Outlier" method, applied to all neurons, achieves a 7.9% performance boost (69.15% vs. 61.25%). This result strongly supports the efficacy of fine-grained mixed-precision quantization. Furthermore, by employing evolutionary algorithms to estimate quantization errors, our optimal quantization configuration allocation method yields an additional 1.34% improvement compared to the standalone "Weight Outlier" allocation method (70.49% vs. 69.15%) .

### 4.3 ABLATION STUDY

**Performance of Neuron Quantization on Advanced Method**. As shown in Table 2, we replicated the expert-level and linear-level approaches proposed by Li et al. (2024) using OmniQuantShao et al. (2023), and compared them with our fine-grained neuron quantization method. The results demonstrate that applying our method at an average bit-width of 2.54 enhances the model's performance to 72.66%. While the relative improvement on the OmniQuant baseline is less pronounced compared to the GPTQ results, it remains statistically significant. This finding underscores the efficacy of our

Table 2: Zero-Shot Task Performance of Mixtral-8x7B using Automated Fine-Grained MoE Quantization with GPTQ

| Model | Bits | Granularity Level | Method | Accuracy (%) ↑ | | | | | |
|---|---|---|---|---|---|---|---|---|---|
| | | | | PIQA | HellaSwag | WinoGrande | OBQA | COPA | Avg. |
| Mixtral-8x7B | fp16 | - | - | 82.54 | 83.99 | 76.32 | 46.80 | 93.00 | 76.53 |
| | 2.54 bit | **Expert** | Frequency | 78.62 | 76.81 | 71.67 | 43.80 | 88.00 | 71.78 |
| | 2.54 bit | **Linear** | Weight Outlier | 79.00 | 77.03 | 71.82 | **45.20** | 89.00 | 72.41 |
| | 2.54 bit | **Neuron (Ours)** | Weight Outlier | **79.05** | **77.05** | **72.22** | 44.00 | **91.00** | **72.66** |

method even when applied to more advanced quantization techniques, thus validating its robustness and generalizability across different quantization frameworks.

## 4.4 CONCLUSION

This paper presents an automated fine-grained quantization method for Mixture-of-Experts (MoE) models, designed to overcome the limitations of current quantization techniques when applied to MoE architectures. As LLMs continue to grow, model compression becomes increasingly important, especially for MoE models with dynamic routing mechanisms. However, the sparsity and heterogeneity of MoE models pose challenges for traditional quantization, which is typically developed for dense models. Current compression methods mainly focus on pruning, such as expert and layer pruning, with relatively little attention given to quantization in MoE models.

To address these challenges, we propose a mixed-precision quantization search space that spans from the expert-level to the neuron-level, allowing for flexible bit-width allocation. This approach adapts to the varying activation patterns and structures in MoE models, improving quantization accuracy. Additionally, we introduce an evolutionary algorithm-based search mechanism to efficiently explore quantization configurations, minimizing quantization loss while adhering to a specified compression rate.

Our experiments show that the proposed method significantly improves MoE model performance across various tasks, particularly in low-bit quantization scenarios. Compared to existing linear quantization techniques, our approach achieves a 7.9% performance improvement at the neuron level, with an additional 1.34% gain through evolutionary algorithms. These results highlight the effectiveness of fine-grained mixed-precision quantization, particularly in optimizing resource allocation for MoE models. Furthermore, our method is orthogonal to existing LLM quantization techniques, enabling potential synergies with other compression methods.

In summary, this work introduces an innovative, automated quantization framework for MoE models that captures their dynamic characteristics and enables precise, granular quantization configurations. By extending quantization to the neuron level, our approach offers new strategies for compressing MoE models and establishes new benchmarks across multiple tasks. This framework holds promise for both academic research and practical deployment of large language models, providing a novel path for efficient model compression.

## 4.5 LIMITATIONS

Despite the promising results, our work has some limitations. Due to resource constraints, we were unable to conduct extensive experiments across a wide range of models. This limits the generalizability of our findings, and further evaluations on different MoE models are necessary to fully validate the effectiveness of our approach. Additionally, current model deployment frameworks do not yet support neuron-level quantization, which restricts the practical application of our method. Future research is needed to develop deployment techniques that can fully exploit the potential of fine-grained, neuron-level quantization, allowing for more efficient real-world implementations of MoE models.

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
