# OpenReview forum: "Automated Fine-Grained Mixture-of-Experts Quantization"
_ICLR.cc/2025/Conference — ICLR 2025 Conference Withdrawn Submission_

### Official Review · Reviewer_Fvoa · 2024-10-31

**Soundness:** 3
**Presentation:** 2
**Contribution:** 2
**Rating:** 5
**Confidence:** 3

**Summary:**

The paper introduces an automated fine-grained quantization framework specifically designed for Mixture-of-Experts (MoE) models, addressing unique challenges these architectures face in quantization due to their inherent sparsity and dynamic routing mechanisms.
The authors propose a mixed-precision quantization search space that ranges from expert-level to channel-level quantization. This approach allows for precise bit-width allocation to different components based on their activation patterns and importance. An evolutionary algorithm is employed to navigate the extensive search space, optimizing quantization configurations and minimizing model loss. Experimental evaluations demonstrate that the proposed method achieves significant performance gains, particularly in low-bit quantization settings, surpassing existing approaches on the Mixtral-8x7b-v0.1 model by 9.24%.

**Strengths:**

The proposed framework not only covers expert-level quantization but also extends to finer levels, such as neuron-level, capturing more nuanced model behaviors.

The authors carefully design their experiments and include extensive zero-shot task performance evaluations that underscore the robustness of their approach, especially in low-bit scenarios.

The method part of this article is detailed and accurate, with good logic, rigorous mathematical formulas and high reproducibility.

**Weaknesses:**

The structure of the article is rather confusing, and the description of previous work should not appear in the method section.

There are no explanatory pictures, which makes the method flow structure unclear.

This paper is not innovative enough and needs to explain how it differs from the existing channel-wise quantization methods obtained through learning.

**Questions:**

What is the difference between evolutionary algorithms and deep learning algorithms that use backpropagation to update parameters?

---

### Official Review · Reviewer_qG6o · 2024-10-31

**Soundness:** 3
**Presentation:** 2
**Contribution:** 2
**Rating:** 5
**Confidence:** 4

**Summary:**

The paper introduced a weight-only quantization approach for Mixture-of-Experts (MoE) based LLMs. The proposed method searched for optimal neuron-level quantization configurations by an evolutionary algorithm. Experimental results showed that the proposed neuron-level quantization outperformed expert-level and layer-level quantization.

**Strengths:**

i) Topic. As a representative reader, the reviewer is interested in reading a paper about fine-grained mixed precision quantization for MoEs. The reviewer believes this topic would be of interest to the ICLR community.

ii) Method. The proposed method is simple yet effective.

iii) Experiment. The paper provided evaluations demonstrating that the neuron-level quantization outperformed expert-level and layer-level quantization.

**Weaknesses:**

i) Implementation. It is unclear to the reviewer if the proposed method would be easy to implement in modern hardware so as to gain real-world inference efficiency. Specifically, the paper proposed to quantize the rows or columns of a weight matrix into different bitwidths, leading to extra heterogeneous operations that may not be easy to gain efficiency in modern GPUs/NPUs. The reviewer wonders if the authors have considered any particular hardware architectures or acceleration techniques that could support such operations efficiently.

ii) Efficiency evaluation. Given the Concern i), it would be better if the authors could provide evaluations on the inference efficiency of the proposed method, compared to vanilla single precision quantization and the baselines included in the paper (i.e. Li 2024). Representative metrics include speed, throughput, peak memory usage, etc.

iii) Performance. In Table 2, with Omniquant, the performance gains of the proposed method looked marginal (< 1%) across all benchmarks. The authors are encouraged to further discuss the statistical significance of these gains and the potential scenarios when the improvements justify the added complexity.

iv) Extra evaluation. Given the limited number of baseline approaches (i.e. only [Li 2024]) presented in the paper, the authors are encouraged to include comparisons to more baselines. One example could be the dense-and-sparse decomposition proposed in SqueezeLLM [1], i.e. separating a tiny set of outliers from the weight matrix and keeping them in full precision. This solution demonstrates strong performance with limited extra bitwidth. It is also simpler than the proposed method.

v) Ablation. The authors are also encouraged to include more ablations, e.g. hyper-parameters for the evolutionary search.

[1] SqueezeLLM: Dense-and-Sparse Quantization. ICML 2024.

**Questions:**

Please refer to Weaknesses

---

### Official Review · Reviewer_Rf2Q · 2024-11-03

**Soundness:** 3
**Presentation:** 3
**Contribution:** 3
**Rating:** 5
**Confidence:** 5

**Summary:**

This paper tackles mixed-precision quantization of Mixture of Expert models. They employ a mixed-precision search space encompassing different granularities from expert-level to channel-level and find the optimal configuration using evolutionary algorithms. The results are encouraging.

**Strengths:**

- MoE quantization is a relevant and useful problem today given the popularity of MoE structures and also the growing problem of resource efficiency at inference setting.
- Authors do a good job covering relevant context for MoE, quantization, and evolutionary algorithms. The method is sound and clear.
- The results support that mixed-precision quantization is helpful for accuracy improvement.

**Weaknesses:**

- For a more comprehensive understanding, It would be better to test a broader range of tasks on top of common sense reasoning tasks that the authors used, such as MMLU and HumanEval. MMLU is known to test for knowledge across a wide range of domains, and HumanEval is a popular metric for coding, which is a popular application area for the open-source community. Thus, adding these benchmarks will strengthen the credibility of model performance.

- As the end result of LLM quantization is to improve inference latency/bandwidth, it would be nice to have some numbers on the measured speedup. This is especially relevant because mixed-granularity methods usually have higher overhead due to more complex operations. For example, comparing the overhead of their mixed-granularity method with more traditional quantization approaches would be helpful.

- Lacks analysis of why this method is superior; needs more specific motivation than "internal structures of experts." Indeed finer granularity is helpful, but the underlying reason is not addressed. Authors could perhaps provide specific examples or hypotheses about how the internal structures of experts contribute to the method's superiority. For instance, I suggest analyzing which aspects of the fine-grained structure are most crucial for performance improvements.

**Questions:**

1. Fine-grained approach can specialize to a specific task, but it is also less generalizable across tasks. It would be interesting to study whether the searched configurations vary between what kind of calibration set is used (target task), and if so, how much variation there exists. Would it be resource-efficient to store many configurations for many tasks? As this uses a GPTQ style approach to find quantization error, it is useful to note that GPTQ also specializes to in-domain calibration sets when doing task-specific quantization, as shown in [1].  To be more specific, authors could quantify the variation in configurations across different tasks and discuss the trade-offs between task-specific optimization and generalization.

2. In sec 2.2, it is fitting to include a recent work (AdaDim [1]) that similarly automates the quantization process using a search algorithm.
3. L85: "internal structures of experts." Please clarify what this means.
4. Adding analysis of what kind of fine-grained structures are exploited would help build intuition for why mixed-precision quantization works. To make it more specific, authors can provide examples or a more detailed explanation of what they mean by "internal structures of experts" in the context of their method. This could include asking how these structures differ from traditional expert-level quantization approaches.

[1] Heo, JH, et al. "Rethinking Channel Dimensions to Isolate Outliers for Low-bit Weight Quantization of Large Language Models." ICLR 2024.

---

### Official Review · Reviewer_iBjP · 2024-11-04

**Soundness:** 2
**Presentation:** 2
**Contribution:** 2
**Rating:** 3
**Confidence:** 5

**Summary:**

This paper proposed a mixed-precision quantization pipeline for a mixture of experts (MoE)
LLMs. The key contribution is to propose an evolutionary algorithm-based approach for identifying optimal quantization configurations to implement flexible expert-level and neuron-level bit-width allocation. Experiments have been conducted in several NLP benchmarks.

**Strengths:**

- The paper's motivation is reasonable. It is a nice attempt to explore the allocation of different bit widths for various experts and neurons within a sparse Mixture of Experts (MoE) architecture.

- The authors have provided the source code, enhancing the paper's clarity and reproducibility, making it easier to follow and re-implement.

**Weaknesses:**

1. In section 3.3, the author selected 2-bit, 2.5-bit, 3-bit, 3.5-bit, and 4-bit as mixed precision alternatives using a fixed ratio method. However, in the experimental section, the author only presented results for 2.54-bit (attention layer 4-bit, expert a total of 2-bit). This lack of comprehensive experimental results renders the final findings insufficient to fully support the article's description and setup.

2. The paper's novelty and contribution are very limited, and it overlooks comparisons with reference methods. As described in Equation (13) of the article, the author's definition of $\varepsilon$ is derived from Equation (6), which is identical to the quantization loss definition in works such as GPTQ and HAWQ (second-order loss). The author adopts this definition and proposes EVOLUTIONARY ALGORITHMS in Algorithm 1 to solve precision. HAWQ employs the same quantization accuracy error and uses Integer Linear Programming (ILP) to solve layer-wise precision. However, in the experimental section, the author only compares a few methods like random, frequency, and weight outlier, without contrasting the Hessian+ILP strategy proposed by HAWQ. This omission raises doubts and makes the experimental results less convincing.

3. The title of the paper is MIXTURE-OF-EXPERTS
QUANTIZATION," but the NEURON QUANTIZATION proposed by the author is not specifically designed for MoE. The down-layer, up-layer, gate-layer FFN structure is widely present in dense LLMs such as LLaMA2. However, the characteristics of the experts manifest in multiple factors like activation frequency and activation weights. The mixed-precision quantization error and strategies designed by the author did not take into account the characteristics of the experts. The author's claim that it is quantization tailored for MoE lacks sufficient evidence.

4. typo: line 310, add \space to 'function' and 'Li et al. (2021)'

**Questions:**

My main concerns lie in the novelty of this paper (as shown in the second point of the Weakness) and insufficient experiments. More details can be found in the weaknesses.

---

### Note · Authors · 2024-11-21

I have read and agree with the venue's withdrawal policy on behalf of myself and my co-authors.